# No trends in spring and autumn phenology during the global warming hiatus

Xufeng Wang [1,2], Jingfeng Xiao [2], Xin Li[3,4], Guodong Cheng [1,5], Mingguo Ma [6], Gaofeng Zhu[7], M. Altaf Arain[8], T. Andrew Black[9] & Rachhpal S. Jassal[9]

Phenology plays a fundamental role in regulating photosynthesis, evapotranspiration, and surface energy fluxes and is sensitive to climate change. The global mean surface air temperature data indicate a global warming hiatus between 1998 and 2012, while its impacts on global phenology remains unclear. Here we use long-term satellite and FLUXNET records to examine phenology trends in the northern hemisphere before and during the warming hiatus. Our results based on the satellite record show that the phenology change rate slowed down during the warming hiatus. The analysis of the long-term FLUXNET measurements, mainly within the warming hiatus, shows that there were no widespread advancing (or delaying) trends in spring (or autumn) phenology. The lack of widespread phenology trends partly led to the lack of widespread trends in spring and autumn carbon fluxes. Our findings have significant implications for understanding the responses of phenology to climate change and the climate-carbon feedbacks.

[1] Key Laboratory of Remote Sensing of Gansu Province, Heihe Remote Sensing Experimental Research Station, Cold and Arid Regions Environmental and Engineering Research Institute, Chinese Academy of Sciences, 730000 Lanzhou, China. [2] Earth Systems Research Center, Institute for the Study of Earth, Oceans, and Space, University of New Hampshire, Durham, NH 03824, USA. [3] Institute of Tibetan Plateau Research, Chinese Academy of Sciences, 100101 Beijing, China. [4] CAS Center for Excellence in Tibetan Plateau Earth Sciences, Chinese Academy of Sciences, Beijing 100101, China. [5] Institute of Urban Study, Shanghai Normal University, 200234 Shanghai, China. [6] School of Geographical Sciences, Southwest University, 400715 Chongqing, China. [7] Key Laboratory of Western China's Environmental Systems (Ministry of Education), Lanzhou University, 730000 Lanzhou, China. [8] School of Geography and Earth Sciences, McMaster University, Hamilton, ON L8S 4K1, Canada. [9] Biometeorology and Soil Physics Group, Faculty of Land and Food Systems, University of British Columbia, Vancouver, BC V6T 1Z4, Canada. Correspondence and requests for materials should be addressed to J.X. (email: j.xiao@unh.edu) or to X.L. (email: xinli@itpcas.ac.cn)

Air temperature is one of the major controlling factors of phenology[1–4], and global warming advances spring phenology (i.e., greenup) and/or delays autumn phenology (i.e., senescence)[5,6]. Phenology has been widely monitored to understand the interactions between vegetation and climate change[7,8]. Changes in spring and autumn phenology can differentially alter the growing season length and affect the carbon, water, and energy fluxes between the terrestrial biosphere and the atmosphere[3,9–11]. Increased carbon uptake resulting from the lengthening of the growing season has the potential to mitigate climate change[12]. Thus, elucidating the trends in phenology can improve our understanding of the impacts of climate change on ecosystem productivity and carbon cycling and the feedbacks to the climate. The global mean surface air temperatures data indicated a global warming hiatus between 1998 and 2012[13–15]. How the warming hiatus affected terrestrial ecosystems has drawn much attention in recent years. A recent study indicated that the warming hiatus led to accelerated net terrestrial carbon uptake in that period, because of reduced respiration[16]. A few studies found the warming hiatus affected the phenology trend[17] using species-level phenology data at the site scale. However, the warming hiatus effects on phenology trends have not been evaluated at the global scale.

The latest FLUXNET dataset provides the first opportunity to examine the trends in spring and autumn phenology for a large number of sites globally using eddy covariance flux data. The eddy covariance technique continuously measures net ecosystem exchange (NEE) at the ecosystem level. The gross primary productivity (GPP) time series partitioned from the NEE observations provides an ecosystem-level means to extract phenology[18,19]. The resulting phenological dates reflect activity of all vegetation within the flux tower footprint (e.g., the extent of the upwind area from which the flux originates). Phenology extracted from eddy covariance carbon flux data has been often used as ground truth to validate phenology estimated from remote-sensing data[19–22]. To our knowledge, however, this FLUXNET database has not yet been used to examine phenological trends over a large spatial domain. Many of the eddy covariance sites have long-term observations and cover the warming hiatus. Thus, the latest FLUXNET dataset can be used to explore phenology trend during the warming hiatus. Meanwhile, the latest GIMMS3g[23] dataset provides long-term normalized difference vegetation index (NDVI) from 1982 to 2015. The NDVI dataset has been widely used to examine phenology for the northern hemisphere or the globe[8,24–26]. Although some of these studies examined the change point in SOS[25,26] or the sensitivity of advancing SOS to global warming[4], how the spring and autumn phenology changed during the warming hiatus compared with the warming period has not been examined using long-term satellite or FLUXNET records. The long duration of the latest GIMMS3g dataset makes it feasible to assess the trends in phenology for both before and during the warming hiatus.

In this study, we used the latest FLUXNET database and the GIMMS3g dataset to examine the trends in spring and autumn phenology in the northern hemisphere and to assess the effects of the warming hiatus on phenology trends. The spring and autumn phenology were estimated from GIMMS3g using five widely used phenology retrieval methods (see "Methods" for details), and the phenology trends before and during the warming hiatus were then compared. We also estimated phenology using the FLUXNET observations and examined the phenology trend during the warming hiatus. Meanwhile, the environmental controls on phenology and the associated changes in carbon fluxes were analyzed based on the FLUXNET database (see "Methods" for the selection of the 56 sites).

## Results

**Phenological trends based on remote sensing data.** The start of growing season (SOS) and end of growing season (EOS) for each grid cell in the northern hemisphere during the period 1982–2014 were estimated from the GIMMS3g dataset using five different methods (see "Methods" for details). To examine the effects of the warming hiatus (1998–2012) on phenology, we examined the phenology trends in the northern hemisphere before and after 1998. The average SOS and EOS estimated using five different methods are shown in Fig. 1 and Table 1. The spring phenology (SOS) significantly advanced before 1998 (slope $= -0.34$ days year$^{-1}$, $p = 0.007$), but had no significant trend after 1998 (slope $= -0.02$ days year$^{-1}$, $p = 0.967$) (Fig. 1a). EOS had a delaying trend before 1998 (slope $= 0.26$ days year$^{-1}$, $p = 0.108$), but had no trend after 1998 (slope $= -0.03$ days year$^{-1}$, $p = 0.902$) (Fig. 1a). Meanwhile, we used five global temperature anomaly datasets (CRUTEM3.0, CRUTEM4.6, NOAA, Berkley Earth, and NASA GISTEMP) to examine the warming trend for spring (Fig. 1c), autumn (Fig. 1d), and the annual scale (Fig. 1e). The rate of the increase in average temperature slowed down after 1998 in spring and at the annual scale. The spatial patterns of GIMMS3g-based phenology and temperature were also compared before and during the warming hiatus (Supplementary Fig. 1). The slowdown of temperature during the warming hiatus was more widespread in North America than in Eurasia. As a result, the lack of advancing trends in SOS and delaying trends in EOS during the warming hiatus was more widespread in North America than in Eurasia.

**Phenological trends based on the FLUXNET database.** Phenological dates were estimated for each site based on the daily GPP data. The average SOS and EOS are shown in Supplementary Fig. 2 for the 56 sites that have at least 7 years of high-quality data. We examined the trends of SOS (Fig. 2a, Supplementary Fig. 3) and EOS (Fig. 2c, Supplementary Fig. 3) for each site using the extracted phenological dates. Among the 56 sites, only five sites had significant SOS trends ($p < 0.05$ at three sites and $0.05 < p < 0.1$ at two sites) with three of them having advancing trends and two of them having delaying trends (Fig. 2b, Supplementary Fig. 3). EOS significantly changed at 12 sites ($p < 0.05$ at 11 sites and $0.05 < p < 0.1$ at 1 site) with 10 of them having delaying trends and 2 of them having advancing trends (Fig. 2d, Supplementary Fig. 3). The spatial distribution of the sites with significant SOS or EOS trends was shown in Supplementary Fig. 4. Our analysis based on the FLUXNET dataset showed that there were no widespread SOS advancing or EOS delaying trends in the northern hemisphere. The measurements of most of the sites fell within the global warming hiatus (Fig. 2a, c), and only 9 sites had measurements before 1998.

**Environmental controls on phenology.** We first performed correlation analysis between environmental factors and phenological dates (SOS and EOS) and then examined the trends of environmental factors. With the auxiliary meteorological data observed at each FLUXNET site, the environmental controls on phenology were explored. The correlation coefficients between phenology and seasonal environmental factors (average air temperature, average shortwave radiation, accumulated precipitation, average VPD, average soil temperature, and average soil water content) were calculated for each of the 56 sites (Supplementary Figs. 5 and 6). The average value of correlation coefficients for the 56 sites is shown in Fig. 3. For SOS, the spring and winter environmental factors were used (Fig. 3a, Supplementary Table 1). For EOS, the summer and autumn environmental factors were used (Fig. 3b, Supplementary Table 2). The

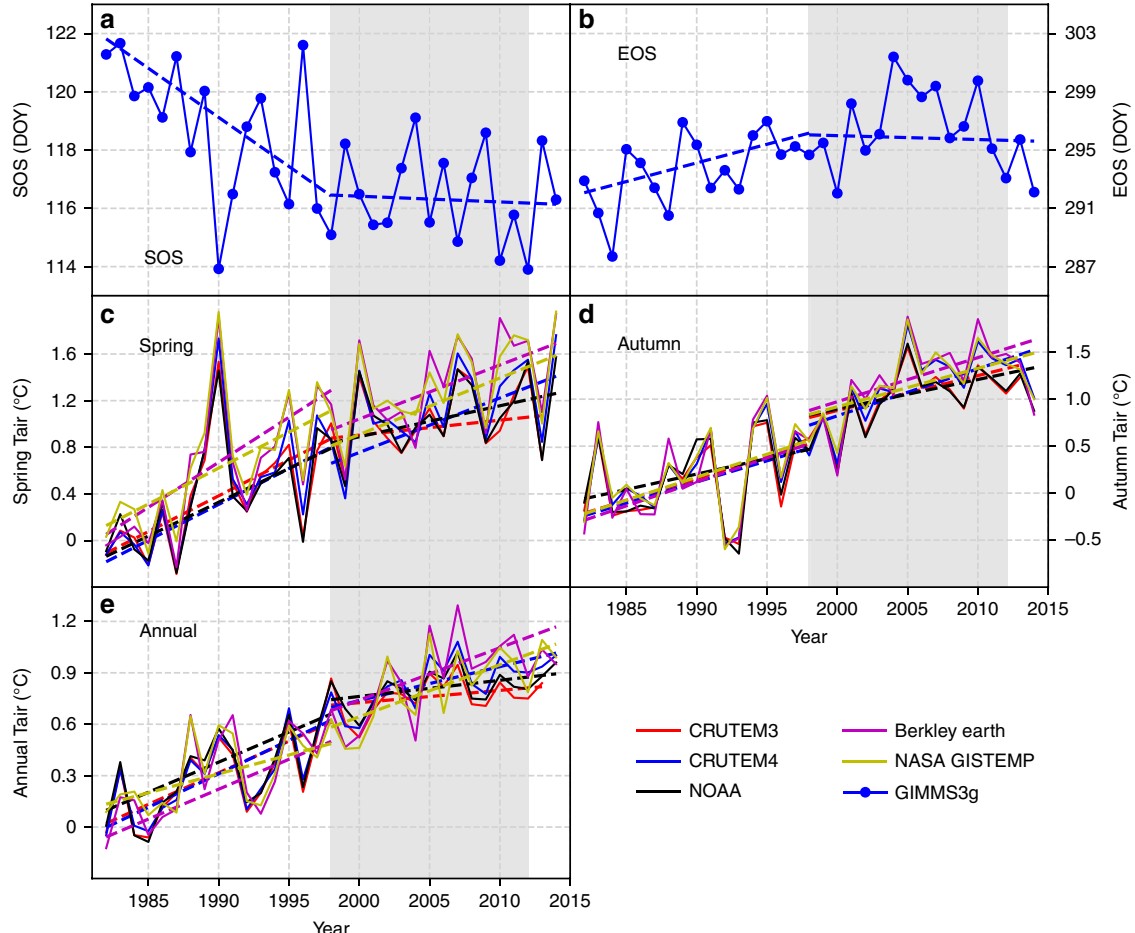

**Fig. 1** Phenology and temperature trends in the northern hemisphere (latitude ≥30°) from 1982 to 2014. **a** The trend of SOS estimated from the GIMMS3g NDVI dataset. **b** The trend of EOS estimated from the GIMMS3g NDVI dataset. **c** The trends of spring temperature anomalies from CRUTEM3.0, CRUTEM4.6, NOAA, Berkley Earth, and NASA GISTEMP. **d** The trends of autumn temperature anomalies from CRUTEM3.0, CRUTEM4.6, NOAA, Berkley Earth, and NASA GISTEMP. **e** The trends of annual temperature anomalies from CRUTEM3.0, CRUTEM4.6, NOAA, Berkley Earth, and NASA GISTEMP. The dashed lines stand for the long-term trend (1982–2014) calculated using the Mann–Kendall Tau-b with Sen's method. The light gray background indicates the warming hiatus (1998–2012). Mann–Kendall trend test was performed for the periods before 1998 and after 1998, separately. SOS start of growing season, EOS end of growing season, DOY day of year

**Table 1 Results of Mann–Kendall trend test performed for phenology and temperature for the periods before 1998 and after 1998**

| Data | | Warming period (1982–1998) | | | Warming hiatus (1998–2014) | | |
|---|---|---|---|---|---|---|---|
| | | Slope | Intercept | *p*-Value | Slope | Intercept | *p*-Value |
| Phenology | GIMMS3g SOS | −0.337 | 789.9 | 0.007 | −0.020 | 155.8 | 0.967 |
| | GIMMS3g EOS | 0.256 | −215.1 | 0.108 | −0.027 | 349.5 | 0.902 |
| Spring Tair | CRUTEM4 | 0.062 | −122.1 | 0.003 | 0.047 | −92.7 | 0.053 |
| | CRUTEM3 | 0.061 | −121.8 | 0.004 | 0.013 | −24.8 | 0.558 |
| | NOAA | 0.058 | −115.0 | 0.009 | 0.027 | −52.3 | 0.149 |
| | Berkley_Earth | 0.077 | −153.1 | 0.003 | 0.046 | −91.7 | 0.077 |
| | NASA-GISTEMP | 0.062 | −122.0 | 0.003 | 0.048 | −96.0 | 0.064 |
| Autumn Tair | CRUTEM4 | 0.045 | −89.3 | 0.077 | 0.050 | −99.4 | 0.015 |
| | CRUTEM3 | 0.046 | −91.5 | 0.064 | 0.037 | −73.4 | 0.017 |
| | NOAA | 0.033 | −65.1 | 0.174 | 0.031 | −61.9 | 0.029 |
| | Berkley_Earth | 0.051 | −101.5 | 0.108 | 0.047 | − 92.8 | 0.023 |
| | NASA-GISTEMP | 0.049 | −97.0 | 0.077 | 0.041 | −80.8 | 0.019 |
| Annual Tair | CRUTEM4 | 0.039 | −77.5 | 0.003 | 0.020 | −38.9 | 0.009 |
| | CRUTEM3 | 0.037 | −73.1 | 0.006 | 0.007 | −13.6 | 0.344 |
| | NOAA | 0.035 | −69.6 | 0.004 | 0.009 | −18.1 | 0.044 |
| | Berkley_Earth | 0.03 | −69.2 | 0.007 | 0.031 | −60.4 | 0.015 |
| | NASA-GISTEMP | 0.022 | −43.6 | 0.044 | 0.030 | −60.0 | 0.004 |

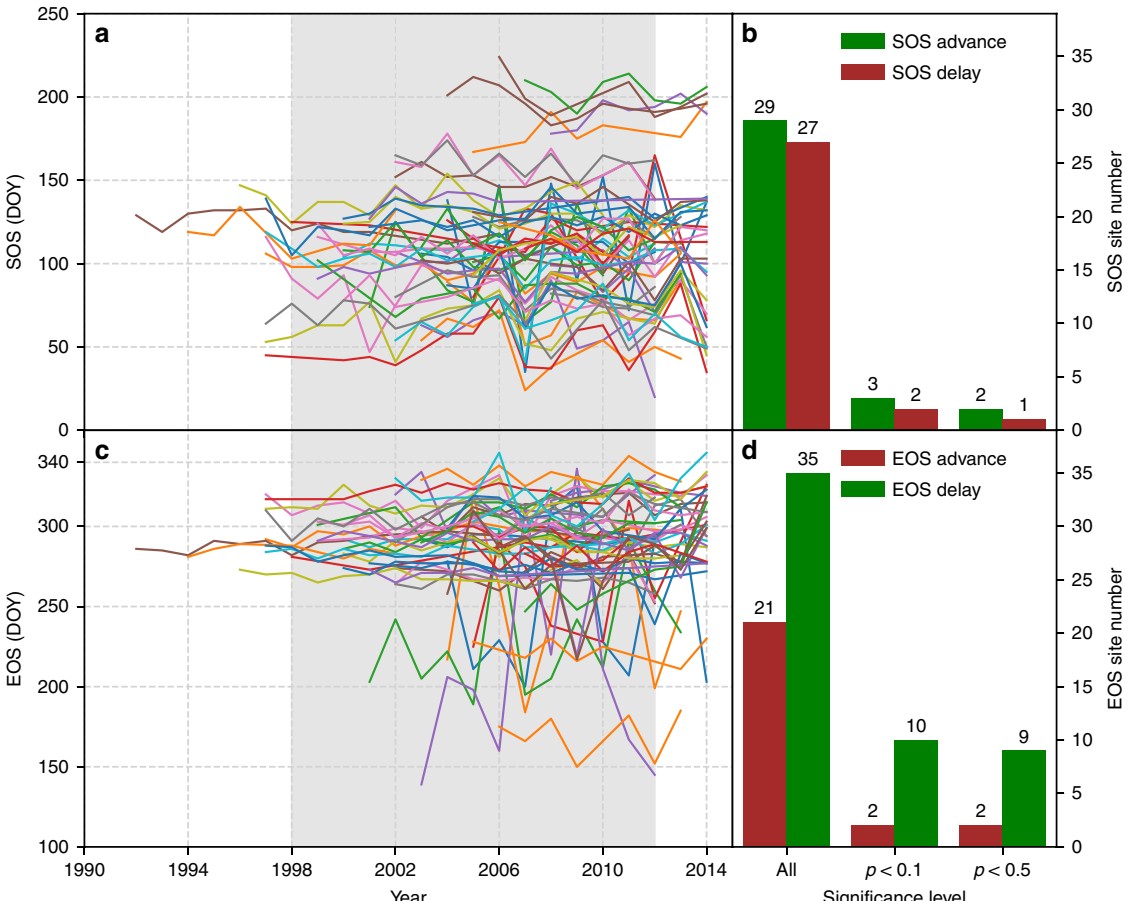

**Fig. 2** Phenology trends at 56 FLUXNET sites with at least 7 years of high-quality data. The left plots are the spaghetti plots of SOS (**a**) and EOS (**c**). The right plots illustrate the number of sites with different SOS (**b**) and EOS (**d**) trends at different significance. Green bars indicate the number of sites with advanced SOS (**b**) or delayed EOS (**d**), while the brown bars indicate the number of sites with delayed SOS (**b**) or advanced EOS (**d**). All stands for the total number of sites with positive slope (or negative slope) regardless of statistical significance. $p < 0.1$ stands for the number of sites with positive slope (or negative slope) that was significant at $p < 0.1$. $p < 0.05$ stands for the number of sites with positive slope (or negative slope) that was significant at $p < 0.05$. The light gray background in **a** and **c** indicate the warming hiatus

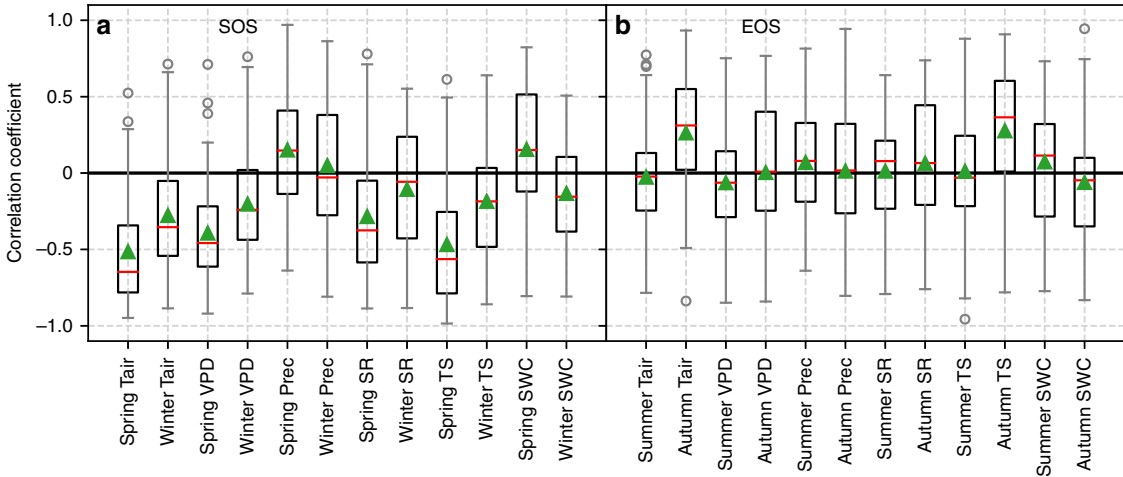

**Fig. 3** Correlation coefficients between phenology (SOS and EOS) and environmental factors at the 56 FLUXNET sites. **a** Correlation coefficient between SOS and environmental factors in spring and winter. **b** Correlation coefficient between EOS and environmental factors in autumn and summer. The environmental factors are as follows: Tair air temperature, VPD vapor pressure deficit, Prec precipitation, SR shortwave downward radiation, TS soil temperature, SWC soil water content. For each box, the central red line is the median, the green triangle is the average, the upper and bottom edge of the box correspond to the 25th and 75th percentiles, the whiskers show the range of the data, and the gray circles are the outliers

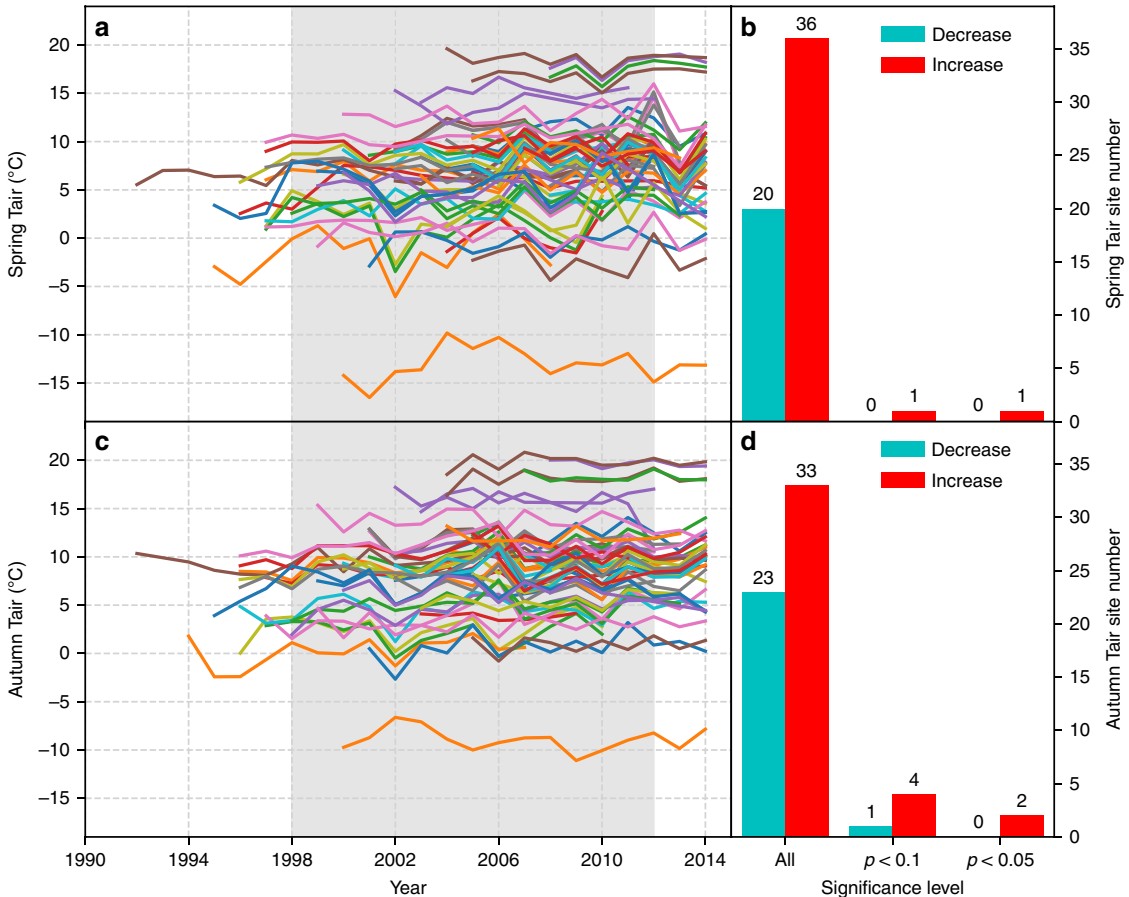

**Fig. 4** Spring and autumn temperature trends at 56 FLUXNET sites. The left plots are the spaghetti plots of spring (**a**) and autumn (**b**) temperature. The right plots are the number of sites with different significance level for spring temperature (**c**) and autumn temperature (**d**). All stands for the total number of sites with positive slope (or negative slope) regardless of significance level. $p < 0.1$ stands for the number of sites with positive slope (or negative slope) that was significant at $p < 0.1$. $p < 0.05$ stands for the number of sites with positive slope (or negative slope) that was significant at $p < 0.05$. The light gray background indicates the warming hiatus

environmental controls on SOS (or EOS) varied greatly among sites (Supplementary Figs. 5 and 6). Generally, SOS showed higher correlation with environmental factors than did EOS. Among the examined environmental factors, spring air temperature had the highest correlation with SOS. At most of the sites, spring air temperature was significantly correlated with SOS (Supplementary Fig. 5). Among the various factors, autumn temperature had the highest correlation with EOS. Moreover, autumn air temperature was significantly correlated with EOS for more sites than was any other environmental variable (Supplementary Fig. 6). The correlation between autumn temperature and EOS was much weaker than that between spring temperature and SOS. To consider the joint controls of multiple environmental factors, the partial correlation analysis was also performed between phenology and environmental factors (Supplementary Fig. 7). Spring temperature had the strongest partial correlation with SOS, and autumn temperature had the strongest partial correlation with EOS. Environmental factors showed stronger control on SOS than on EOS. These results were generally similar to those based on the univariate statistical analyses.

We then examined the trends for environmental factors of both SOS (Supplementary Fig. 8) and EOS (Supplementary Fig. 9). At almost all the five sites with significant SOS trends, at least one environmental factor had a significant trend and was also significantly correlated with SOS (more detailed analysis is given in Supplementary Note 1). Our study showed that spring air temperature was the main controlling factor of SOS at these sites,

and the spring air or soil temperature was negatively correlated with SOS. This is consistent with many previous studies[27–29]. By examining the trend, however, spring air temperature significantly increased at only one site (Fig. 4a). Other factors played minor roles in the changes in SOS.

Similarly, for most of the 12 sites with significant EOS trends, at least one of the meteorological variables that were significantly correlated with EOS had a significant trend (more detailed analysis is given in Supplementary Note 1). Autumn air temperature was the major controlling factor of EOS (Fig. 3b). For most of the sites without EOS trends, there was no significant trend in any of the meteorological variables that were significantly correlated with EOS. There were also few sites that the EOS trend was decoupled with meteorological variables.

**Impacts of changes in phenology on carbon fluxes**. To examine the effects of the trends in SOS (or EOS) on carbon fluxes, we calculated the correlation coefficients between SOS and spring carbon fluxes (Supplementary Fig. 10), and between EOS and autumn carbon fluxes (Supplementary Fig. 11), as well as the Mann–Kendall trend of carbon fluxes at each site (Supplementary Fig. 12). Changes in spring and/or autumn phenology can alter ecosystem carbon fluxes. An advancing trend in SOS resulted in increasing trends in carbon fluxes at three of five sites with significant SOS trends. Similarly, a delaying trend in EOS resulted in increasing trends in carbon fluxes at most of the 12 sites with significant EOS trends (more detailed analysis is also given in

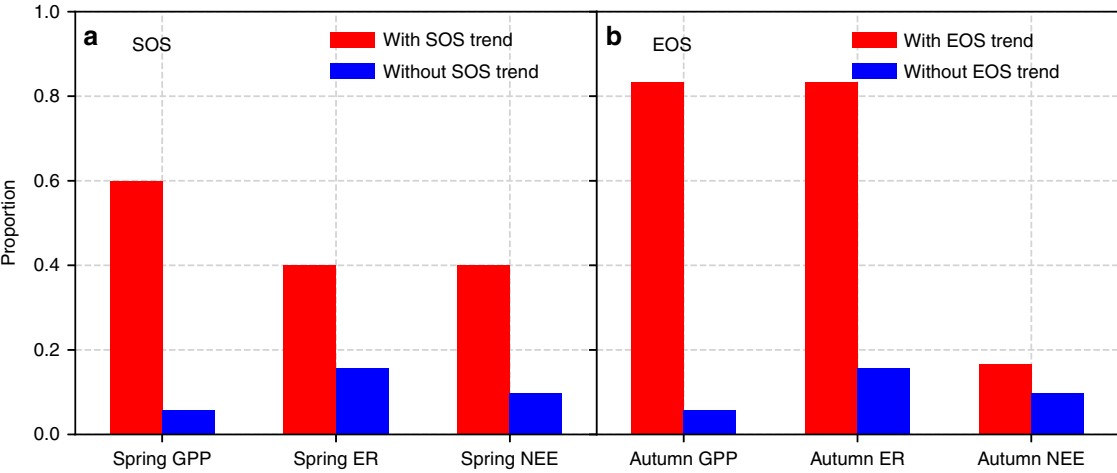

**Fig. 5** Proportion of sites with significant carbon fluxes trends among sites with and without SOS or EOS trends. In **a**, the red and blue bars stand for the proportion of sites with and without significant SOS trend, respectively; in **b**, the red and blue bars stand for the proportion of sites with and without significant EOS trend, respectively. SOS start of growing season, EOS end of growing season

Supplementary Note 2). The proportion of sites with significant GPP and ecosystem respiration (ER) trends was higher for the sites with significant SOS (or EOS) trends than for the sites without significant SOS (or EOS) trends (Fig. 5). Due to the offset between GPP and ER, SOS (or EOS) trends had weaker impact on NEE trends compared with GPP and ER. GPP had the strongest correlation with SOS (or EOS), followed by ER and NEE (Supplementary Tables 3 and 4). Advancing SOS could strengthen GPP, ER, and NEE in spring. Delaying EOS could also strengthen GPP, ER, and NEE. EOS had weaker effects on NEE than SOS.

## Discussion

Based on long-term satellite records, our results showed that there was no significant trend in SOS or EOS during the warming hiatus. A previous study using long-term data on flower onset date at three sites in Germany found that the flowering advancing rate largely decreased during the warming hiatus[17]. In our study, the northern hemisphere SOS estimated from remote-sensing data stopped advancing and the EOS exhibited an insignificant trend during the warming hiatus. Generally, the trends in spring and autumn phenology were consistent with those in spring and autumn temperature, respectively. Many studies based on satellite records showed advancing SOS (or delaying EOS)[30], and one recent study indicated that the SOS advancing (or EOS delaying) rate during the period 2002–2012 was even faster than that during the period 1982–1992[31]. Another study indicated that although the SOS advancing slowed down in some parts of the northern hemisphere, it was unlikely to slow down over the whole northern hemisphere over the period 1982–2011[25]. Our study found that SOS (or EOS) stopped advancing (or delaying) during the warming hiatus, and this is different from the findings of most previous studies. The discrepancies in phenology trends among studies are due to the differences in satellite records, data processing procedures, and phenology extraction methods as well as the associated uncertainties[30]. Our study focused on the phenology trend during the warming hiatus (1998–2012) which was not previously examined. Compared with previous studies, we also used the latest GIMMS NDVI3g dataset. We divided the entire study period into two shorter periods according to the definition of the warming hiatus. We performed a sensitivity test to examine the effects of the start and end years for the hiatus on phenology trends because the trend analysis for a short period might be sensitive to the start (or end) values[25]. The sensitivity test showed that the SOS (or EOS) based on the satellite record

stopped advancing (or delaying) during the warming hiatus regardless of the start and/or end years (Supplementary Fig. 13). The GIMMS3g-based phenology also contains uncertainties due to data quality issues caused by noise, sensor degradation, and calibration between different sensors[32].

Our analysis based on the FLUXNET database showed that more sites (12) had significant trends in autumn phenology than in spring phenology. By averaging the SOS (or EOS) changing rate among all these sites, EOS had higher change rate than SOS. This is inconsistent with most of the remote sensing-based studies showing that the SOS advancing rate was much higher than the EOS delaying rate[8]. Previous ground-based phenological observations also indicated that the EOS delaying rate was much faster than the SOS advancing rate at some sites[12,33]. The phenology trends at three of the 56 sites were also examined by a previous study[3], and their results are consistent with our work at these sites.

Extracting phenology trends from eddy covariance flux data has advantages over other phenology monitoring methods. First, ecosystem-scale SOS and EOS can be reliably determined from tower GPP data. Remote sensing-based phenology usually contains large uncertainties because vegetation indices like NDVI are more of proxies of vegetation greenness and also have significant uncertainty resulting from various sources (e.g., cloud contamination, snow cover, cross-sensor calibration, and sensor degradation)[22]. Second, the carbon flux-based phenology is useful for understanding phenology dynamics at the ecosystem level. Third, the flux towers measure NEE, surface energy exchange, evapotranspiration (ET), and meteorological variables, and these data along with phenological estimates can be used to analyze the impacts of phenological changes on the carbon, energy, and water fluxes at the ecosystem scale. Finally, eddy covariance flux sites are distributed across the globe and the observing methods are standardized. Among the 914 registered FLUXNET sites, 191 sites have more than 10 years of observations. Should more flux tower data be shared with the research community in the future, the flux tower data could be used to examine changes in phenology for a larger number of sites globally. Detecting phenology from carbon flux data also faces some challenges. First, flux data contain uncertainties due to random measurement error[34], and these measurement error can result in uncertainties in phenology estimation. Second, for some sites, there are substantial data gaps in the shoulder seasons due to instrument malfunction or bad data quality, which hinders the correct estimation of phenological

stages. Third, the gap-filling and partitioning of NEE could lead to uncertainty in GPP, which will in turn introduce uncertainty to the estimates of phenological dates. Finally, although there are a large number of flux sites over the globe encompassing a range of climate and ecosystem types, the resulting phenology estimates are not spatially continuous.

Our results from the GIMMS NDVI3g and the FLUXNET database were generally consistent with each other. The GIMMS NDVI3g dataset-based phenology showed no trends in SOS (or EOS) during the warming hiatus. The phenology based on the FLUXNET GPP data showed no widespread SOS (or EOS) trends in the northern hemisphere during the warming hiatus. By extracting the SOS (or EOS) from the GIMMS NDVI3g phenology estimates at the FLUXNET sites, the satellite-based SOS (or EOS) trends at the FLUXNET sites were also examined. The trends of average satellite-based SOS (or EOS) at the FLUXNET sites were consistent with those averaged for the northern hemisphere. Before the warming hiatus, the average satellite-based SOS at FLUXNET sites significantly advanced but no trend was observed during the warming hiatus (Supplementary Fig. 14). The average satellite-based EOS at the FLUXNET sites had no trend either before or during the warming hiatus (Supplementary Fig. 14).

Many previous studies reported that spring temperature was the main driver of phenology changes, and increasing spring temperature led to advanced SOS[28,29]. In some temperate ecosystems, winter temperature increase could result in delayed SOS because the chilling requirements were not met[6,35]. By contrast, our results showed that winter air temperature and winter soil temperature were also negatively correlated with SOS for most of the flux sites. Previous studies indicated that SOS trends can result from trends in either spring or winter temperature. For example, warmer spring temperature may result in the advancing in SOS[5,36], while some other studies attributed the reversed SOS trend to warming winter and the subsequent failure of the chilling requirement[37,38]. Our results showed that SOS was more strongly correlated with spring temperature than with winter temperature, and the lack of trend in spring temperature was mainly responsible for the lack of SOS trend at most sites. For some sites, SOS change was decoupled with climatic factors and was mainly controlled by other factors (more detailed analysis is also given in Supplementary Note 1). In particular, SOS at cropland sites could be controlled by human management, such as rotation, sowing, and harvesting, to a large extent. For example, all climatic factors (air temperature, soil temperature, VPD, precipitation, and shortwave radiation) had very weak correlation with SOS, but the sowing date had a strong relationship with SOS at a cropland site in Nebraska, USA (US-Ne2) (Supplementary Fig. 15). Some previous studies showed that EOS was controlled by temperature, particularly the late summer or autumn temperature[39,40], while some other studies indicated that EOS was weakly correlated with air temperature[41]. Precipitation and solar radiation were also main determinants of EOS at some ecosystems[24]. Across all 56 sites in our study, autumn air and soil temperature had larger effects on EOS than did other factors.

Compared to EOS, SOS was generally more sensitive to environmental factors. EOS may also be impacted by vegetation development during the growing season or some climate events. Previous studies mainly documented the effects of temperature on phenology, and other environmental factors were not well studied[9]. We found that, in general, temperature, VPD, and shortwave radiation were the main factors influencing the interannual changes in phenology across these sites, although precipitation and soil water content had larger impacts on phenology than other factors for a few sites. The environmental effects on phenology varied with site. Vegetation phenology is jointly controlled by a combination of many environmental factors and the status of vegetation. The correlation between a single environmental factor and phenology was often relatively weak and the consistency between the trend of a single environmental factor and the trend of spring or autumn phenology was relatively poor for many sites. Meanwhile, the footprint of many flux sites is spatially heterogeneous with different plant species. To examine whether the species composition affected the revealing of the underlying mechanisms of phenological responses, we grouped the forest sites into two groups based on the site description information: one group with one dominant species and one group with multiple dominant species. The comparison of the relationships between phenology and environmental variables between these two groups indicated that the species composition had little impact on the environmental controls on phenology, especially SOS (Supplementary Fig. 16).

A significant trend in SOS (or EOS) can result in significant trends in spring (or autumn) GPP and RECO (more detailed analysis is also given in Supplementary Note 2). For the majority of the 56 sites, however, there were no significant trends in spring (or autumn) carbon fluxes because of the insignificant trends in phenology during the warming hiatus. Some previous studies reported accelerated global land carbon uptake during the warming hiatus[16,42]. Based on FLUXNET data at the selected 56 sites, our study showed that few sites had significant trends in carbon fluxes (GPP, ER, or NEE) despite insignificant trends in phenology likely because of increased vegetation cover or enhanced photosynthesis. For the majority of the 56 sites, however, there were no trends in spring/autumn carbon fluxes because of the lack of significant trends in SOS (or EOS) phenology; the correlation of annual carbon fluxes with SOS (or EOS) were also very weak (Supplementary Tables 3 and 4). The increase in net carbon uptake during the warming hiatus in recent studies has been attributed to reduced ER[16], lower land-use emissions caused by decreased tropical forest loss and temperate afforestation[42], and enhanced peak growth[43]. Our findings along with these recent studies indicated that the enhanced net carbon uptake during the warming hiatus was likely not due to the lengthening of the growing season but to other factors (e.g., reduced ER, land use change, and enhanced peak growth). Moreover, the time period of our analysis is not exactly the same as those of the previous studies. Uncertainties in carbon fluxes caused by random measurement error[34], gap-filling method[44], and NEE partitioning method[45,46] may result in uncertainties in the carbon flux trends and the relationship between phenology and seasonal carbon fluxes.

In conclusion, based on phenology estimated from a long-term satellite record using different methods, our study showed that the phenology change rate in the northern hemisphere, especially the SOS advancing rate, slowed down during the warming hiatus. Meanwhile, our study showed that among the 56 FLUXNET sites in the northern hemisphere with most observations within the warming hiatus, significant trends in spring and autumn phenology were observed for only 5 and 12 sites, respectively, and most sites had no significant trends in either spring or autumn phenology. Environmental effects on phenology varied among different sites. Generally, among the environmental factors examined, air temperature in spring and autumn were significantly correlated with SOS and EOS, respectively. With the FLUXNET records mainly falling in the warming hiatus, only very few sites had significant increasing trends in air temperature in spring and autumn. At most of the sites where phenology did not significantly change, no significant trend was found for any environmental factor. The trends in spring (or autumn) carbon fluxes depended on SOS (or EOS) trends, and there were no widespread increasing trends in spring (or autumn) carbon fluxes

partly because of the lack of widespread trends in SOS (or EOS). Our findings have implications for understanding the impacts of climate change on vegetation phenology, the influences of phenological changes on carbon uptake and vegetation productivity, and thereby the climate–carbon cycle feedbacks. Rising air temperatures driven by the buildup of carbon dioxide and other greenhouse gases can advance SOS and/or delay EOS and thereby lead to the lengthening of the growing season. A longer growing season can increase plant productivity and ecosystem carbon uptake which will in turn partly offset carbon emissions, and a longer growing season can also enhance transpiration and potentially reduce soil water availability and runoff. Our findings show that the slowdown of climatic warming during the warming hiatus led to the stabilization of spring and autumn phenology. This indicates that the stabilization of climate in the future will likely stabilize phenology and growing season length. With the stabilization of phenology, ecosystems would not have an additional carbon uptake period; meanwhile, ecosystems would be able to maintain the lengths of the different seasons and thereby the seasonality, and would not increase water loss via enhanced transpiration. Some recent studies, however, indicated that regional or global mean temperatures may continue to increase following zero carbon emissions[47,48], although a number of studies showed that global mean surface temperatures would stay roughly constant for a couple of centuries after carbon emissions are stopped[49,50]. It also remains unclear when zero carbon emissions will be achieved. Therefore, the stabilization of phenology globally is likely not to be anticipated for the foreseeable future unless another warming hiatus occurs. Our findings can improve our understanding of the responses of phenology to the warming hiatus and the carbon–climate feedbacks.

## Methods

**Datasets**. The latest GIMMS NDVI3g (GIMMS3g) NDVI dataset was used to estimate phenology for the northern hemisphere[23]. The data was available for the period from 1982 to 2015. The latest global flux database, FLUXNET2015, was first released in December 2015 with more site–years of data added in November 2016 (http://fluxnet.fluxdata.org/data/fluxnet2015-dataset/)[51]. This dataset consists of 212 sites across the globe, and these sites encompass 13 major vegetation types (Supplementary Fig. 17). The eddy covariance data for all the sites were processed with the same procedures to minimize inconsistency and uncertainties caused by data processing for cross-site synthesis. Compared with its predecessor—the 2007 LaThuile database, the new FLUXNET2015 database has been improved in both data quality control and data processing procedures. To explore the long-term trends in phenology, we identified and used the sites having at least 7 years of high-quality carbon flux data (≥75% of good quality data in a year). One exception is DK-ZaH, where the percentage of good-quality GPP data was higher than 0.75 during the growing season that was <2 months long and was very low in the non-growing season. For some evergreen broadleaf forests, GPP does not exhibit obvious seasonality, and we excluded these sites from this study. Finally, 56 sites met our criteria and all these sites are located in North America and Europe (Supplementary Fig. 17). The latitude of the 56 sites ranged from 31.74 (US-Wkg) to 74.47°N (DK-ZaH). More detailed information on these sites is given in Supplementary Data 1, including site name, site ID, vegetation type, site coordinates (latitude/longitude), start year of measurement, end year of measurement, duration, number of years with high-quality data and references. Seasonal carbon fluxes and meteorological variables were calculated from monthly data to explore the impacts and controlling factors of phenology. Seasonal carbon fluxes or meteorological variables were excluded if the percentage of good-quality data was smaller than 75%. The global monthly temperature anomaly datasets used in this study were provided by Climatic Research Unit, University of East Anglia (CRUTEM4[52]; CRUTEM3[53]), NASA Goddard Institute for Space Studies[54], NOAA National Centers for Environmental Information[55], and Berkeley Earth[56]. These are the major temperature datasets used in the IPCC report to assess the global warming rate[57]. Spring, autumn, and annual mean temperature anomalies were calculated for the northern hemisphere. All the datasets contain the whole warming hiatus.

**NDVI-based phenology estimation**. There are a variety of methods to extract phenology dates from the NDVI time series. Generally, they can be divided into two groups. The first is inflection point detection method, which finds the inflection point in the smoothed annual NDVI time series curve. The date when the derivative of the smoothed NDVI is the local maxima (or minima) is used as the SOS in spring or the EOS in autumn. The second group is threshold method,

which compares the smoothed NDVI time series with a fixed NDVI value (i.e., fixed threshold method) or a fixed percentage of the annual maximum NDVI (i.e., dynamic threshold method because the threshold NDVI value changed inter-annually). Here, we selected five widely used phenology estimating methods with two inflection point detection methods and three threshold methods to determine the SOS and EOS for the northern hemisphere.

Method 1: First, a double logistic function (Eq. (1)) was fitted with the time series NDVI, and then the second-order derivative of the fitted curve was calculated. The two dates corresponding to the two local maxima points in the first half year are the SOS and the onset of maturity. The two dates corresponding to the two local maxima points in the second half year are the onset of senescence and the EOS[58,59].

$$y(t) = a + b\left(\frac{1}{1 + e^{c(t-d)}} + \frac{1}{1 + e^{e(t-f)}}\right) \qquad (1)$$

where $c$, $d$, $e$, and $b$ are parameters of this function, $a$ is the initial background NDVI value, $a + b$ is the maximum NDVI value, $t$ is time in days, and $y(t)$ is the NDVI value at time $t$.

Method 2: First, a double logistic function (Eq. (1)) was fitted with the time series NDVI, then the date when NDVI increasing (or decreasing) fastest was used as SOS (or EOS). Specifically, the date corresponding to the maxima (or minima) value in first-order derivative of the fitted curve was determined as SOS (or EOS)[60].

Methods 3 and 4: The dynamic threshold NDVI was used to extract phenology. The NDVI was fitted with a double logistic function. Then, the fitted NDVI was normalized using the following function:
$\text{Ratio}_{day} = (\text{NDVI}_{day} - \text{NDVI}_{min})/(\text{NDVI}_{max} - \text{NDVI}_{min})$, where $\text{NDVI}_{day}$ is fitted NDVI at given day, $\text{NDVI}_{max}$ and $\text{NDVI}_{min}$ are maximum and minimum NDVI each year. A threshold ratio was used to determine SOS and EOS in spring and autumn. Here, we used the threshold 0.2[37] in method 3 and 0.5[61] in method 4.

Method 5: A fixed threshold NDVI value was used to determine the phenology. The NDVI season dynamic was calculated from the multiyear NDVI, then the NDVI change rate ($\text{NDVI}_{RC}$) was estimated from the multi-year averaged NDVI seasonal dynamics using the following function: $\text{NDVI}_{RC} = \frac{\text{NDVI}(t+1) - \text{NDVI}(t)}{\text{NDVI}(t)}$, where $\text{NDVI}(t+1)$ and $\text{NDVI}(t)$ are NDVI value at time $t+1$ and $t$, respectively; The NDVI values correspond to maximum and minimum NDVI change rate are used as the thresholds for SOS and EOS, respectively. The first and second half year NDVI was fitted with a polynomial function: $\text{NDVI} = a + a1 \times t + a2 \times t^2 + \cdots + an \times t^n$, $n = 6$, where $t$ is Julian day in a year, and $a$, $a1$, $a2$, …, $a6$ are fitting parameters. SOS and EOS were determined by comparing the fitted daily NDVI with the thresholds[62,63].

We selected the northern hemisphere with latitude greater than 30° to estimate phenology because vegetation has evident seasonality in this area. The area with annual mean NDVI <0.1 was considered as non-vegetated area and was excluded in remote sensing-based phenology estimation. The Savizky–Golay filter was applied to the GIMMS NDVI3g dataset to minimize the noise before phenology estimation. The parameter setting for Savizky–Golay is the same as a previous paper[64].

**Phenology estimation from the FLUXNET database**. A moving average filter was used to smooth the daily GPP curve for each of the 56 FLUXNET sites. Within the moving window, the weights were calculated from the Gaussian density function in which the weight is the highest for the mid-point of the time window and decreases to the two sides. The dates for SOS and EOS were then extracted from the smoothed GPP curve based on a GPP threshold. The SOS was determined as the date when the smoothed daily GPP was greater than the GPP threshold in spring, while the EOS was determined as the date when the smoothed daily GPP was less than the GPP threshold in autumn (Supplementary Fig. 18). In this study, we used 15% of the multi-year daily GPP maximum value as our threshold following previous studies[65,66], and the resulting threshold value varied between 1 and 2 g C m$^{-2}$ day$^{-1}$ for different ecosystems. By changing the threshold from 4% to 30% with a step of 2%, we found the standard deviation of estimated phenological dates from different NEE partitioning methods became stable when the threshold was greater than 15% (Supplementary Fig. 19). The FLUXNET dataset provided GPP estimates based on both respiration extrapolation approach[45] and light use efficiency approach[46] We compared the SOS (EOS) extracted from both GPP estimates, and found that they were almost identical with each other (Supplementary Fig. 20). We therefore used GPP based on the respiration extrapolation approach.

**Phenology trends and correlates**. The Mann–Kendall[67,68] method was used to examine the trends in phenology and meteorological variables. Because the Mann–Kendall method is a nonparametric test for monotonic trends, it does not assume a specific distribution for the data and is insensitive to outliers. Because of these advantages, it has been widely used for trend analysis recently. The Theil–Sen method was used to estimate the slope of the Mann–Kendall trend[69]. We also analyzed the environmental controls on phenology and impacts of phenology on ecosystem carbon exchange. For each environmental factor (air temperature, soil temperature, VPD, precipitation, soil moisture, or shortwave radiation) and carbon flux (GPP, NEE, or ER), the seasonal and annual values were calculated from monthly values. The seasons are defined as spring (March–May), summer

(June–August), autumn (September–November), and winter (December of the previous year to February). Seasonal meteorological data were quality controlled by filtering low-quality data (QC flag < 75%). Then the correlation and partial correlation coefficients were calculated between phenological dates and meteorological variables at both seasonal and annual scales. The trends of these meteorological variables for different seasons were also analyzed using the Mann–Kendall method to examine whether there were consistent trends between environmental factors and phenology. Correlation analysis was also performed between carbon fluxes and phenology to examine the contribution of the changes in phenology to inter-annual variations of carbon fluxes.

## Data availability

Source data for Figs. 1–5 and Supplementary Figs. 1–16 are provided as a Source Data file and can be accessed from https://github.com/XufengWangofCAS/NCOMMS-SourceDataFile. The gridded SOS and EOS data (1982–2014) retrieved from the GIMMS NDVI3g dataset using five different retrieval methods are available for downloading at http://globalecology.unh.edu. The GIMMS NDVI3g version1 was provided by the NASA Ames Ecological Forecasting Lab (https://ecocast.arc.nasa.gov/data/pub/gimms/). The FLUXNET2015 database was downloaded from http://fluxnet.fluxdata.org/data/ fluxnet2015-dataset/. The temperature anomaly was provided by Climatic Research Unit, University of East Anglia (RUTEM4: https://crudata.uea.ac.uk/cru/data/temperature/CRUTEM4-gl.dat and CRUTEM3: https://crudata.uea.ac.uk/cru/data/crutem3/CRUTEM3-gl.dat), NASA Goddard Institute for Space Studies (https://data.giss.nasa.gov/pub/gistemp/GHCNv3/gistemp250.nc.gz), the NOAA National Centers for Environmental Information (ftp://ftp.ncdc.noaa.gov/pub/data/ghcn/v3/grid/grid-mntp-1880-current-v3.3.0.dat.gz) and Berkley Earth (http://berkeleyearth.lbl.gov/auto/Global/Gridded/Complete_TAVG_LatLong1.nc). More raw data in this study are available from the corresponding author upon request.

## Code availability

The code used to estimate phenology from remote sensing data and carbon flux can be accessed from https://github.com/XufengWangofCAS/NCOMMS-Codes. More codes in this study are available from the corresponding author upon request.

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

## Acknowledgements

This work was jointly supported by China's funding agencies (Strategic Priority Research Program of the Chinese Academy of Sciences: Grant No. XDA19070204 and XDA20060600, the National Natural Science Foundation of China: Grant nos. 41771466 and 91425303 and the Special Fund for Key Program of Science and Technology of Qinghai Province: Grant no. 2017-SF-A6) and U.S. funding agencies (National Aeronautics and Space Administration (NASA) through the Carbon Cycle Science Program: Grant No. NNX14AJ18G and the Climate Indicators and Data Products for Future National Climate Assessments: NNX16AG61G and National Science Foundation through MacroSystems Biology Program: Grant No. 1638688). We thank all the PIs and other research personnel of the FLUXNET sites for making the flux data available. This work used eddy covariance data acquired and shared by the FLUXNET community, including these networks: AmeriFlux, AfriFlux, AsiaFlux, CarboAfrica, CarboEuropeIP, CarboItaly, CarboMont, ChinaFlux, Fluxnet-Canada, Global Water Futures Program, GreenGrass, ICOS, KoFlux, LBA, NECC, OzFlux-TERN, TCOS-Siberia, and USCCC. The ERA-Interim reanalysis data are provided by ECMWF and processed by LSCE. The FLUXNET eddy covariance data processing and harmonization was carried out by the European Fluxes Database Cluster, AmeriFlux Management Project, and Fluxdata project of FLUXNET, with the support of CDIAC and ICOS Ecosystem Thematic Center, and the OzFlux, ChinaFlux, and AsiaFlux offices. We thank the three anonymous reviewers for their constructive and insightful comments on the manuscript.

## Author contributions

X.F.W. designed research, performed research, analyzed data, and wrote the paper; J.F.X. designed research, performed research, analyzed data and wrote the paper; X.L. designed research and wrote the paper; M.A.A. provided eddy covariance data and provided comments and suggestions on the manuscript; T.A.B. provided eddy covariance data and provided comments and suggestions on the manuscript; R.S.J. provided eddy covariance data and provided comments and suggestions on the manuscript; M.G.M. provided valuable discussions; G.F.Z. provided valuable discussions; G.D.C. provided valuable discussions. All authors contributed to the revisions of the manuscript.

## Additional information

**Competing interests:** The authors declare no competing interests.

