## [Peer Review File · Nature Communications]

Reviewers' comments:

Reviewer #1 (Remarks to the Author):

This study evaluates the long-term changes in phenology during the last 3 decades. The main finding in this study is SOS advancing trend has been weakened during the recent decade. But, this is not a new finding, there are several studies to address this “change” in long-term phenology trends. Significant earlier spring phenology trends by the end of 1990s has been weakened after 2000s across the Northern Hemisphere (Piao et al., 2011; Zeng et al., 2011; Fu et al., 2014,,,,,,,,). In terms of mechanism, there are several studies to explain observed phenomena. For example, Fu et al mentioned changes in spring phenology sensitivity to explain changes in trends of spring phenology. In addition, we know strong linkage between temperature and phenology as well. Then, what is the main novelty in this study? Authors should check previous studies carefully.

Wang, X., Piao, S., Xu, X., Ciais, P., MacBean, N., Myneni, R.B., Li, L., 2015. Has the advancing onset of spring vegetation green-up slowed down or changed abruptly over the last three decades? *Global Ecology and Biogeography* 24, 621–631. <https://doi.org/10.1111/geb.12289>

Fu, Y.H., Piao, S., Op de Beeck, M., Cong, N., Zhao, H., Zhang, Y., Menzel, A., Janssens, I.A., 2014a. Recent spring phenology shifts in western Central Europe based on multiscale observations. *Global Ecology and Biogeography* 23, 1255–1263. <https://doi.org/10.1111/geb.12210>

Fu, Y.H., Piao, S., Zhao, H., Jeong, S.-J., Wang, X., Vitisse, Y., Ciais, P., Janssens, I.A., 2014b. Unexpected role of winter precipitation in determining heat requirement for spring vegetation green-up at northern middle and high latitudes. *Glob Change Biol* 20, 3743–3755. <https://doi.org/10.1111/gcb.12610>

Fu, Y.H., Zhao, H., Piao, S., Peaucelle, M., Peng, S., Zhou, G., Ciais, P., Huang, M., Menzel, A., Peñuelas, J., Song, Y., Vitisse, Y., Zeng, Z., Janssens, I.A., 2015. Declining global warming effects on the phenology of spring leaf unfolding. *Nature* 526, 104–107. <https://doi.org/10.1038/nature15402>

Reviewer #2 (Remarks to the Author):

Review: No trends in spring and autumn phenology during the global warming hiatus period (see attachment)

Wang et al. submitted to Nature Communications

Wang et al. analysed long-term remote sensing data and FLUXNET data in order to investigate phenological trends before and after the hiatus. Their results are already summarized in the title: No trends in spring and autumn phenology during the global warming hiatus period.

The paper is easy to understand and clearly written. The study addresses a larger scale, the subject is timely and of interest for a wide public. However, there are some issues that have to be addressed (major revision):

It is correctly stated in L 272 that “vegetation phenology is jointly controlled by a combination of many environmental factors...” However, the authors only investigated single meteorological factors in univariate statistical analyses.

In addition, numerous plant species with different (temperature) responses contribute to the FLUXNET GPP measurement which makes it difficult to obtain mechanistic explanations.

In addition, intrinsic uncertainties in ecosystem fluxes should be discussed as well.

The authors conclude that “Our findings have important implications for understanding the impacts of climate change on vegetation phenology, the influences of phenological changes on carbon uptake and vegetation productivity, and thereby the climate-carbon cycle feedbacks.” Given the heterogeneous results and the above mentioned issues, it is hard for the reader to imagine the detailed implications. Please specify.

Maybe it is worth to test the sensitivity of the results to the start and end dates for the hiatus period: For example, the pronounced El Niño event of 1997/98 strongly affected temperature (and probably GPP estimates).

The limitations arising from the separation of the two periods for calculating two distinct estimates / trends has to be discussed in more detail. More mechanistic explanations for the variability or trends in phenological data, e.g. related to the effect of a reduced chilling, etc., would strengthen the study.

Although being described as a global phenomenon, the hiatus is linked to regionally distinct patterns in temperature trend changes. In turn, resolving these patterns of the interplay between climate and vegetation is necessary to interpret the global picture.

Minor remarks

L 30: Phenology is sensitive to temperature and therefore to global warming.

L 56: It is worth to note that the effect of an advancing spring has the highest influence on the length on the vegetation period. A (slight) prolongation of autumn is of minor importance.

L 76: specify "tower footprint"

Reviewers' comments:

Reviewer #1 (Remarks to the Author):

This study evaluates the long-term changes in phenology during the last 3 decades. The main finding in this study is SOS advancing trend has been weakened during the recent decade. But, this is not a new finding, there are several studies to address this “change” in long-term phenology trends. Significant earlier spring phenology trends by the end of 1990s has been weakened after 2000s across the Northern Hemisphere (Piao et al., 2011; Zeng et al., 2011; Fu et al., 2014,,,,,,). In terms of mechanism, there are several studies to explain observed phenomena. For example, Fu et al mentioned changes in spring phenology sensitivity to explain changes in trends of spring phenology. In addition, we know strong linkage between temperature and phenology as well. Then, what is the main novelty in this study? Authors should check previous studies carefully.

Wang, X., Piao, S., Xu, X., Ciais, P., MacBean, N., Myneni, R.B., Li, L., 2015. Has the advancing onset of spring vegetation green-up slowed down or changed abruptly over the last three decades? *Global Ecology and Biogeography* 24, 621–631. <https://doi.org/10.1111/geb.12289>

Fu, Y.H., Piao, S., Op de Beeck, M., Cong, N., Zhao, H., Zhang, Y., Menzel, A., Janssens, I.A., 2014a. Recent spring phenology shifts in western Central Europe based on multiscale observations. *Global Ecology and Biogeography* 23, 1255–1263. <https://doi.org/10.1111/geb.12210>

Fu, Y.H., Piao, S., Zhao, H., Jeong, S.-J., Wang, X., Vitasse, Y., Ciais, P., Janssens, I.A., 2014b. Unexpected role of winter precipitation in determining heat requirement for spring vegetation green-up at northern middle and high latitudes. *Glob Change Biol* 20, 3743–3755. <https://doi.org/10.1111/gcb.12610>

Fu, Y.H., Zhao, H., Piao, S., Peaucelle, M., Peng, S., Zhou, G., Ciais, P., Huang, M., Menzel, A., Peñuelas, J., Song, Y., Vitasse, Y., Zeng, Z., Janssens, I.A., 2015. Declining global warming effects on the phenology of spring leaf unfolding. *Nature* 526, 104–107. <https://doi.org/10.1038/nature15402>

Response: We thank the reviewer for the comment on our manuscript. We have carefully read all the references that the reviewer suggested. Although some of these studies were already cited in the original manuscript, we have further revised the manuscript (Line 87-90; Line 131-145; Line 279) in order to include others (reference numbers are 10, 26, 27 and 33 in the revised manuscript).

Wang et al. (2015) used three satellite datasets (AVHRR, Terra-MODIS and SPOT) to test the SOS trends over the period 1982-2011 for the land within 30°–75° N latitude in the Northern Hemisphere, and concluded that SOS advance is unlikely to have slowed down or changed abruptly at a hemispheric scale over the last three decades. Only in some parts of the Northern Hemisphere, SOS advance could have slowed down or abruptly changed.

Fu et al. (2014a) used *in situ* leaf unfolding observations of 31 plants and NDVI data at the corresponding sites to test the phenology trend over the period 1982-2011 in Western Central Europe, and found that both of them indicated spring phenology significantly advanced during the period 1982-2011. But for the period 2000-2011, the spring phenology advance rate significantly weakened, and opposite spring phenology trends were found between *in situ* and NDVI. The *in situ* observations indicated a slower but still advancing trend for leaf unfolding, whereas the NDVI series showed a delayed spring phenology during 2000-2011. They thought the recent trend reversal in the advancement of spring phenology in western Central Europe was related to the response of early spring species to the cooling trend in late winter.

Fu et al. (2014b) used CRU-NCEP v4 climate data set and NDVI based green-up date during the period 1982-2009 in 30°–75° N latitude in the Northern Hemisphere to explore the effect of the heat requirement (growing degree days: GDD) on vegetation green-up date, and showed that GDD requirement, chilling and precipitation may have complex interactions in their effects on spring vegetation green-up phenology. They found previous winter season precipitation is positively correlated with growing degree days (GDD), which is widely used to assess the effect of temperature on plant development.

Fu et al. (2015) used long-term *in situ* observations of leaf unfolding for seven dominant European tree species and CRU temperature data from 1980 to 2013 to test the leaf unfolding temperature sensitivity. They found that leaf unfolding temperature sensitivity declined from 4.0 ± 1.8 days °C⁻¹ during 1980–1994 to 2.3 ± 1.6 days °C⁻¹ during 1999–2013.

Compared with these previous studies, our work has following novelties: (1) To our knowledge, this is the first study to explicitly examine the effects of the warming hiatus on phenology for the northern hemisphere; (2) The recently released FLUXNET dataset makes it possible to examine the phenology trend globally and is used in this study to examine the phenology trends for the entire northern middle and high latitudes for the first time; (3) The FLUXNET dataset allows us to explore the environmental controls on phenology and the impacts of phenological changes on carbon fluxes; and (4) We combined both remote sensing and FLUXNET datasets and these two independent, long-term records agreed that the phenology trends were insignificant during the warming hiatus period. This makes our study novel and unique. Therefore, we believe that despite previous relevant studies, our work is of sufficient novelty.

Reviewer #2 (Remarks to the Author):

Wang et al. analysed long-term remote sensing data and FLUXNET data in order to investigate phenological trends before and after the hiatus. Their results are already summarized in the title: No trends in spring and autumn phenology during the global warming hiatus period.

The paper is easy to understand and clearly written. The study addresses a larger scale, the subject is timely and of interest for a wide public. However, there are some issues that have to be addressed (major revision):
Response: We thank the reviewer for the positive evaluation of our manuscript. We have fully considered the specific points raised by the reviewer and revised the manuscript accordingly.

It is correctly stated in L 272 that “vegetation phenology is jointly controlled by a combination of many environmental factors...” However, the authors only investigated single meteorological factors in univariate statistical analyses.

Response: In the revision, we have added partial correlation analysis between phenology and environmental factors to consider the joint controls of environmental factors on phenology. We chose temperature, precipitation, downward shortwave radiation and vapor pressure deficit in spring and winter as explanatory variables. We performed partial correlation between SOS and these environmental variables in spring and winter and partial correlation between EOS and these environmental variables in summer and autumn. We used the ppcor package in R environment to do the partial correlation analysis. Spring temperature had the strongest partial correlation with SOS; autumn temperature has the strongest partial correlation with EOS; environmental factors showed stronger control on SOS than on EOS. These results were generally similar to those based on the univariate statistical analyses. We have added the partial correlation analysis in the main text (Line 258-265) and the supporting information (Line 188; Line 355-363), and also added a figure in the supporting information (labeled as Figure S13).

In addition, numerous plant species with different (temperature) responses contribute to the FLUXNET GPP measurement which makes it difficult to obtain mechanistic explanations.

Response: We agree that the presence of more than one plant species at a site makes it more complicated to explore the underlying mechanisms. To examine whether the species composition can affect the revealing of the underlying mechanisms of phenological responses, we grouped the forest sites into two groups based on the site description information: one group consisting of sites with one dominant species or planted forest and the other group consisting of sites with more than one dominant species. Then we compared the

phenology-environment relationships between these two groups. The average correlation coefficients between phenology and environmental factors for group 1 (4 sites) and group 2 (12 sites) are shown in Fig. S17 in the Supporting Information. The comparison shows that the species composition had little impact on the environmental controls on phenology, especially SOS. In the revision, we have described the effects of species composition in the main text (Line 324-332), and included a new figure in the Supporting Information (Figure S17). For other vegetation types, we did not have sufficient species composition information.

In addition, intrinsic uncertainties in ecosystem fluxes should be discussed as well.

Response: As suggested, we have added discussion about the intrinsic uncertainties in ecosystem fluxes. Please see Line 218-224; Line 384-387.

The authors conclude that “Our findings have important implications for understanding the impacts of climate change on vegetation phenology, the influences of phenological changes on carbon uptake and vegetation productivity, and thereby the climate-carbon cycle feedbacks.” Given the heterogeneous results and the above mentioned issues, it is hard for the reader to imagine the detailed implications. Please specify.

Response: As suggested, we have specified the implications as follows: “Rising air temperatures driven by the buildup of carbon dioxide and other greenhouse gases can advance SOS and/or delay SOS and thereby lead to the lengthening of the growing season. A longer growing season can increase plant productivity and ecosystem carbon uptake which will in turn partly offset carbon emissions, and a longer growing season can also enhance transpiration and potentially reduce soil water availability and water yield. Our findings show that the slowdown of climatic warming during the warming hiatus period led to the stabilization of spring and autumn phenology. This indicates that the stabilization of climate in the future as a result of carbon mitigation efforts will likely stabilize phenology and growing season length. With the stabilization of phenology, ecosystems would not be able to have an additional carbon uptake period; meanwhile, ecosystems would be able to maintain the lengths of the different seasons and thereby the seasonality, and would not increase water loss via enhanced transpiration. Understanding the responses of phenology to the warming hiatus is therefore of scientific and political importance.” (Lines 406-420)

Maybe it is worth to test the sensitivity of the results to the start and end dates for the hiatus period: For example, the pronounced El Niño event of 1997/98 strongly affected temperature (and probably GPP estimates).

Response: In the revision, using the SOS and EOS estimated from GIMMS, and spring, autumn and annual temperatures, we performed a sensitivity test to examine the effects of the start and end years for the hiatus period on phenology trends. The sensitivity test showed that the SOS (or EOS) stopped advancing (or delaying) during the warming hiatus regardless of the start and/or end years. In the revision, we have discussed this sensitivity test in the main text (Lines 146-151) and included a new figure in the Supporting Information (labeled as Figure S6).

The limitations arising from the separation of the two periods for calculating two distinct estimates / trends has to be discussed in more detail.

Response: We have examined and discussed the potential effects of separating the study period into two periods as mentioned above in the response to the previous comment.

More mechanistic explanations for the variability or trends in phenological data, e.g. related to the effect of a reduced chilling, etc., would strengthen the study.

Response: As suggested by the reviewer, we have revised the discussion by adding more mechanistic explanations for the trends in phenological data as follows: “Previous studies indicated that SOS trends can result from trends in either spring or winter temperature. For example, warmer spring temperature may result in the advance in SOS (Wang et al., 2011; Menzel et al., 2006), while some other studies attributed the reversed SOS trend to warming winter and the subsequent failure of the chilling requirement (Yu et al., 2010; Pope et al., 2013). In our study, SOS was more strongly correlated with spring temperature than with winter temperature, while the lack of trend in spring temperature was mainly responsible for the lack of SOS trend at most sites.” (Lines 287-293)

Although being described as a global phenomenon, the hiatus is linked to regionally distinct patterns in temperature trend changes. In turn, resolving these patterns of the interplay between climate and vegetation is necessary to interpret the global picture.

Response: In the revision, we compared the spatial patterns of GIMMS3g based phenology and CRUTEM4 temperature trend (as shown in Fig. S5). The slowdown of temperature during the warming hiatus was more widespread in North America than in Eurasia. As a result, the lack of advancing trends in SOS and delaying trends in EOS during the warming hiatus period was more widespread in North America than in Eurasia. We have included this discussion in the main text (Lines 120-125) and a new figure in the Supporting Information (Figure S5).

Minor remarks

L 30: Phenology is sensitive to temperature and therefore to global warming.

Response: We have revised this sentence as suggested (Line 30).

L 56: It is worth to note that the effect of an advancing spring has the highest influence on the length on the vegetation period. A (slight) prolongation of autumn is of minor importance.

Response: As suggested, we have revised this sentence (Line 58).

L 76: specify “tower footprint”

Response: We have specified “tower footprint” by modifying the sentence as follows: “The resulting phenological dates reflect activity of all vegetation within the flux tower footprint (e.g., the extent of the upwind area from which the flux originates).” (Lines 77-78).

Reviewer #3 (Remarks to the Author):

In *No trends in spring and autumn phenology during the global warming hiatus period* Wang et al. use the so-called warming hiatus as a natural experiment to test the response of the global C cycle. To investigate if there was any phenological response during this period Wang et al. look at two key C cycle datasets- the long-term satellite NDVI record and a global compilation of Fluxnet sites. While the lack of significant trends in the NDVI data are compelling the Fluxnet data are not really suitable for testing any changes in the trend due to their shorter duration. Furthermore the fluxnet data analysis and results presented need to be placed in the context of previous research. Overall, I think that this paper adds an interesting component to the ever-growing work looking at changes in the global C cycle during the warming hiatus and I think that it should be published after some minor revisions.

Response: We thank the reviewer for the positive evaluation of our manuscript. Compared with the GIMMS3g dataset, the duration of FLUXNET2015 dataset is indeed shorter, and the short duration may result in uncertainty in trend analysis. That is why we only chose sites with no fewer than 7 years of

good-quality measurements. Among these selected sites, many of them had valid data of more than 10 years, and some sites had even up to 18 years of data. Thus, the trend analysis of these selected sites was reliable and the FLUXNET data provide new and independent evidence for the impacts of the warming hiatus on phenology.

General Comments:

Changes in temperature trends and satellite NDVI trends largely reflect what has been shown previously (Keenan et al. 2016; Ballantyne et al. 2017), but Wang et al. break down these records into seasons, which is a new contribution. The Fluxnet data are less suitable for testing hypotheses related to phenological responses to the warming hiatus because there is so little data prior to 1998- the putative starting date of the hiatus. The fact that the authors conclude no trends in phenology has to be placed in the context of previous research, in particular ref. 8 Keenan et al. that concluded from flux sites that net C uptake has increased due to phenological response to warming. So are the handful of sites that show a significant trend in EOS (n=11) included in the Keenan study?

Response: We appreciate the reviewer's observation on Keenan et al. that net C uptake has increased due to phenological response to warming. Their study only focused on temperate forest in eastern US with 7 eddy covariance sites (5 DBF sites and 2 ENF sites) for data up to 2010. Of these 7 sites, only 3 sites (US-MMS, US-UMB and US-Ha1) are included in the FLUXNET Dataset2015. For these 3 sites, our results are consistent with Keenan's results (supporting information in Keenan et al. (2014)) that SOS (or EOS) at US-Ha1 was found significantly advanced (or delayed) and SOS (or EOS) have no significant trend at US-MMS and US-UMB). We have now mentioned this in the revised manuscript (Line 199-201).

It would also be nice to know where these sites were located and what biomes they represent.

Response: We mapped the sites with significant phenology trends and used different symbols to differentiate different biome types. In this revision, we have described this in the main text (Line 177-178) and added this map into the Supporting Information (Figure S9).

If in fact, there are no trends in phenology but there is evidence of increased net C uptake from the eddy flux data this must mean that the growing season is not getting appreciably longer, just more productive. I think that the authors in general need to revisit the literature in their conclusions to help reconcile their results with previous research.

Response: We appreciate the reviewer's comment. In the revision, we have revisited the literature and reconciled our results with previous findings as follows: "Some previous studies reported accelerated global land carbon uptake during the warming hiatus period^{17,43}. Based on FLUXNET data at the selected 56 sites, our study showed that few sites with insignificant trends in phenology had significant trends in carbon fluxes (GPP, RECO, or NEE) likely because of increased vegetation cover or enhanced photosynthesis. For the majority of the 56 sites, however, there were no trends in spring (or autumn) carbon fluxes because of the lack of significant trends in SOS (or EOS) phenology; the correlation of annual carbon fluxes with SOS (or EOS) were also very weak (Table S4 and S5). The increase in net carbon uptake during the warming hiatus period in recent studies has been attributed to reduced ER¹⁷, lower land-use emissions caused by decreased tropical forest loss and temperate afforestation⁴³, and enhanced peak growth⁴⁴. Our findings along with these recent studies indicated that the enhanced net carbon uptake during the warming hiatus period was not due to the lengthening of the growing season but to other factors (e.g., reduced ER, land use change, and enhanced peak growth). Moreover, the time period of our analysis is not exactly the same as those of the previous studies." (Line 369-384)

Specific Comments:

L43 only five sites experienced advancing trends in spring and only 12 sites experienced delaying trends in fall.

Response: We have corrected this to “only three sites experienced advancing trends in spring, and 10 sites experienced delaying trends in autumn” (Line 42-44).

L68 Reference 17 seems to be about reptiles and amphibians and not directly about vegetation. There are countless studies on changes in vegetation phenology, so you probably don't need to cite this study that is less relevant to vegetation

Response: We have deleted this reference as suggested.

L96 Move '(See Methods...)' to end of paragraph

Response: We have revised this as suggested (Line 102-103).

L113 Seems odd that they only examined two datasets both generated by the climate research unit. Although most of these gridded surface temperature records share meteorological sites, there are more independently derived datasets from NASA, NOAA, Berkely Earth. Why did you pick these two datasets that may have similar biases.

Response: According to the reviewer's suggestion, we have downloaded the other three temperature datasets from NOAA, NASA and Berkely Earth. We then checked the temperature trend before and during the warming hiatus using these three additional datasets. Our new analysis showed that the warming rate in spring and autumn (particularly in spring) of these three datasets also slowed down during the warming hiatus. This is consistent with the finding of Medhaug et al. (2017). Medhaug et al. (2017) compared global GRU, NOAA, BEST, Berkely Earth and Cowtan & Way surface temperature datasets from 1980 to 2015, and the temperature increasing speed slowdown was clearly seen in all these datasets during 1998-2012. We have now stated this in the revised manuscript (Line 116-117); we also included the three new datasets in Figure 1 and added a new table (Table 1) to summarize the slope and p-values for the trends in NDVI and the four temperature datasets in the manuscript.

L 125 'the trends in spring and fall phenology'

Fig. 1 There have been problems with sensor degradation on some of the satellites that go into the GIMMS3g dataset (Jiang et al. 2017) that may affect changes in the trend presented by the authors. Most of the figure captions appear to have strange underscore notation ' _ '.

Response: We have rephrased the sentence (Line 131-132 in manuscript), and have added discussion about the uncertainties in phenology estimation caused by sensor degradation (Line 151-152 in manuscript). We have also corrected all figure captions as suggested.

L 156 Did any of these 9 sites indicate a change in slope prior to and during the hiatus?

Response: Of these 9 sites, 6 started in 1997, one in 1996, one in 1994 and one in 1992. The valid data time series is too short to examine the trends for both prior to and during the hiatus.

L 170 SOS/EOS is this the ratio, or are you simply referring to SOS and EOS throughout.

Response: SOS/EOS indicates SOS or EOS throughout the manuscript. We have corrected it as “SOS (or EOS) ” throughout the manuscript and Supporting Information to avoid confusion.

L 195 What method for estimating GPP did you use- respiration extrapolation approach (Reichstein et al. 2005) or the light use efficiency approach (Lasslop et al. 2010) often times both methods are reported.

Response: The FLUXNET dataset provided GPP estimates based on both respiration extrapolation approach

(Reichstein et al. 2005) and light use efficiency approach (Lasslop et al. 2010). We compared the SOS (EOS) extracted from both GPP estimates partitioned with these two methods, and found that they were almost identical with each other (see Figure S4). We therefore opted to use the more widely used method - respiration extrapolation approach. In the revision, we have described this in the Methods section (Lines 484-488), and included a new figure in the Supporting information (listed as Figure S4).

L212 Odd first sentence considering that no significant phenology trends were reported. Fig. 3 why are no error bars or standard deviations reported on these average coefficients.

Response: We have rephrased the first sentence as suggested (Line 241 in the manuscript). We have also revised Fig. 3 by using boxplots.

L 257 where were these sites located? And what biomes do they represent? Perhaps they correspond with some regions in the high arctic that appeared to warm during the hiatus.

Response: As mentioned above, we have mapped the distribution of these sites and used different symbols to differentiate biomes. In this revision, we have described this in the main text (Line 177-178) and added this map into the Supporting Information (Figure S9).

L 259 Fig. 3b?

Response: We have corrected this error (Line 303 in the manuscript).

L304 make sure that GPP and RECO are defined on first-use.

Response: We have defined both GPP (Line 75 in the manuscript) and RECO on their first-use (Line 352 in the manuscript). In addition, we have replaced RECO with ER as ER is more widely used now.

L 307 - 309 Seemingly contradictory results 'autumn fluxes higher correlations with EOS...EOS had weaker effects on NEE.'

Response: We have corrected the sentence (Line 353-355 in the manuscript).

L324 photosynthesis rate has increased significantly

Response: We have revised the sentence (Line 373 in the manuscript).

L344 It is not totally clear what these implications are after having read the discussion. Earth's terrestrial ecosystems continue to take up more C each year, so does it matter when this C is taken up- maybe?

Response: Reviewer #2 raised the same comment. In the revision, we have explicitly specified the implications as follows: "Rising air temperatures driven by the buildup of carbon dioxide and other greenhouse gases can advance SOS and/or delay SOS and thereby lead to the lengthening of the growing season. A longer growing season can increase plant productivity and ecosystem carbon uptake which will in turn partly offset carbon emissions, and a longer growing season can also enhance transpiration and potentially reduce soil water availability and water yield. Our findings show that the slowdown of climatic warming during the warming hiatus period led to the stabilization of spring and autumn phenology. This indicates that the stabilization of climate in the future as a result of carbon mitigation efforts will likely stabilize phenology and growing season length. With the stabilization of phenology, ecosystems would not be able to have an additional carbon uptake period; meanwhile, ecosystems would be able to maintain the lengths of the different seasons and thereby the seasonality, and would not increase water loss via enhanced transpiration. Understanding the responses of phenology to the warming hiatus is therefore of scientific and political importance." (Lines 406-420)

References:

- Ballantyne, A., W. Smith, W. Anderegg, and P. Kauppi. 2017. "Accelerating Net Terrestrial Carbon Uptake during the Warming Hiatus due to Reduced Respiration." *Nature Climate Change*.
- Balzarolo M, Vicca S, Nguy-Robertson AL, Bonal D, Elbers JA, Fu YH, et al. Matching the phenology of Net Ecosystem Exchange and vegetation indices estimated with MODIS and FLUXNET in-situ observations. *Remote Sens Environ* 2016, 174: 290-300.
- Fu YH, Zhao H, Piao S, Peaucelle M, Peng S, Zhou G, et al. Declining global warming effects on the phenology of spring leaf unfolding. *Nature* 2015, **526**(7571): 104.
- Fu YH, Piao S, Op de Beeck M, Cong N, Zhao H, Zhang Y, et al. Recent spring phenology shifts in western Central Europe based on multiscale observations. *Global Ecology and Biogeography* 2014a, **23**(11): 1255-1263.
- Fu YH, Piao S, Zhao H, Jeong S-J, Wang X, Vitasse Y, et al. Unexpected role of winter precipitation in determining heat requirement for spring vegetation green-up at northern middle and high latitudes. *Global Change Biol* 2014b, 20(12): 3743-3755.
- Fu YH, Piao S, Delpierre N, Hao F, Hänninen H, Liu Y, et al. Larger temperature response of autumn leaf senescence than spring leaf - out phenology. *Global Change Biol* 2018, **24**(5): 2159-2168.
- Fyfe JC, Meehl GA, England MH, Mann ME, Santer BD, Flato GM, et al. Making sense of the early-2000s warming slowdown. *Nature Climate Change* 2016, 6: 224.
- Huang K, Xia J, Wang Y, Ahlström A, Chen J, Cook RB, et al. Enhanced peak growth of global vegetation and its key mechanisms. *Nature Ecology & Evolution* 2018, **2**(12): 1897-1905.
- Jiang, Chongya, Youngryel Ryu, Hongliang Fang, Ranga Myneni, Martin Claverie, and Zaichun Zhu. 2017. "Inconsistencies of Interannual Variability and Trends in Long-Term Satellite Leaf Area Index Products." *Global Change Biology* 23 (10): 4133 - 46.
- Jeong SJ, Ho CH, Gim HJ, Brown ME. Phenology shifts at start vs. end of growing season in temperate vegetation over the Northern Hemisphere for the period 1982-2008. *Global Change Biol* 2011, 17(7): 2385-2399.
- Keenan TF, Gray J, Friedl MA, Toomey M, Bohrer G, Hollinger DY, et al. Net carbon uptake has increased through warming-induced changes in temperate forest phenology. *Nature Climate Change* 2014, **4**(7): 598-604.
- Keenan, Trevor F., I. Colin Prentice, Josep G. Canadell, Christopher A. Williams, Han Wang, Michael Raupach, and G. James Collatz. 2016. "Recent Pause in the Growth Rate of Atmospheric CO₂ due to Enhanced Terrestrial Carbon Uptake." *Nature Communications* 7 (November): 13428.
- Lasslop, Gitta, Markus Reichstein, Dario Papale, Andrew D. Richardson, Almut Arneth, Alan Barr, Paul Stoy, and Georg Wohlfahrt. 2010. "Separation of Net Ecosystem Exchange into Assimilation and Respiration Using a Light Response Curve Approach: Critical Issues and Global Evaluation." *Global Change Biology* 16 (1): 187 - 208.
- Medhaug I, Stolpe MB, Fischer EM, Knutti R. Reconciling controversies about the 'global warming hiatus'. *Nature* 2017, **545**: 41.
- Menzel A, Sparks TH, Estrella N, Koch E, Aasa A, Ahas R, et al. European phenological response to climate change matches the warming pattern. *Global Change Biol* 2006, **12**(10): 1969-1976.
- Piao S, Huang M, Liu Z, Wang X, Ciais P, Canadell JG, et al. Lower land-use emissions responsible for increased net land carbon sink during the slow warming period. *Nat Geosci* 2018, **11**(10): 739-743.
- Pope KS, Dose V, Da Silva D, Brown PH, Leslie CA, DeJong TM. Detecting nonlinear response of spring phenology to climate change by Bayesian analysis. *Global Change Biol* 2013, 19(5): 1518-1525.
- Reichstein, Markus, Eva Falge, Dennis Baldocchi, Dario Papale, Marc Aubinet, Paul Berbigier, Christian Bernhofer, et al. 2005. "On the Separation of Net Ecosystem Exchange into Assimilation and Ecosystem Respiration: Review and Improved Algorithm." *Global Change Biology* 11 (9): 1424 - 39.
- Wang S, Yang B, Yang Q, Lu L, Wang X, Peng Y. Temporal Trends and Spatial Variability of Vegetation Phenology over the Northern Hemisphere during 1982-2012. *Plos One* 2016, 11(6): e0157134.

- Wang X, Piao S, Ciais P, Li J, Friedlingstein P, Koven C, et al. Spring temperature change and its implication in the change of vegetation growth in North America from 1982 to 2006. *Proceedings of the National Academy of Sciences* 2011, 108(4): 1240-1245.
- Wang, X., Piao, S., Xu, X., Ciais, P., MacBean, N., Myneni, R.B., Li, L., Has the advancing onset of spring vegetation green-up slowed down or changed abruptly over the last three decades? *Global Ecology and Biogeography*, 2015, 24, 621–631.
- Wu C, Peng D, Soudani K, Siebicke L, Gough CM, Arain MA, *et al.* Land surface phenology derived from normalized difference vegetation index (NDVI) at global FLUXNET sites. *Agricultural and Forest Meteorology* 2017, **233**: 171-182.
- Yu H, Luedeling E, Xu J. Winter and spring warming result in delayed spring phenology on the Tibetan Plateau. *Proceedings of the National Academy of Sciences* 2010, 107(51): 22151-22156
- Zeng H, Jia G, Epstein H. Recent changes in phenology over the northern high latitudes detected from multi-satellite data. *Environ Res Lett* 2011, 6(4): 045508.

REVIEWERS' COMMENTS:

Reviewer #1 (Remarks to the Author):

no more comments, would like to accept.

Reviewer #2 (Remarks to the Author):

The manuscript written by Wang et al. improved a lot and they adressed all the concerns that I raised at the first stage of the review process. I think that this manuscript is now suitable for publication in Nature Communications since it makes a valuable contribution in this field of research. I am only wondering if we might face a stabilization of climate in the future as a result of carbon mitigation efforts and thus a stabilization of phenology and growing season length. I stronly argue that this has to be rewritten since temperature will most likly continue to increase in the future.

Reviewer #3 (Remarks to the Author):

I approve of the revisions and authors responses to my previous concerns.

I did find this error in the final paragraph though: 'Rising air temperatures driven by the buildup of carbon dioxide and other greenhouse gases can advance SOS and/or delay SOS and thereby lead to the lengthening of the growing season'

I think that this should read 'SOS and/or delay of EOS' because it does not make sense currently.

ash

Reviewer #1 (Remarks to the Author):

no more comments, would like to accept.

Response: We sincerely thank Reviewer #1 for the time and effort in reviewing our manuscript.

Reviewer #2 (Remarks to the Author):

The manuscript written by Wang et al. improved a lot and they addressed all the concerns that I raised at the first stage of the review process. I think that this manuscript is now suitable for publication in Nature Communications since it makes a valuable contribution in this field of research. I am only wondering if we might face a stabilization of climate in the future as a result of carbon mitigation efforts and thus a stabilization of phenology and growing season length. I strongly argue that this has to be rewritten since temperature will most likely continue to increase in the future.

Response: We sincerely thank Reviewer #2 for the time and effort in reviewing our manuscript. We agree with the reviewer that temperatures will likely continue to increase in the future (Lines 821-838). Therefore, we have modified this part as follows:

“Our findings show that the slowdown of climatic warming during the warming hiatus period led to the stabilization of spring and autumn phenology. This indicates that the stabilization of climate in the future will likely stabilize phenology and growing season length. With the stabilization of phenology, ecosystems would not have an additional carbon uptake period; meanwhile, ecosystems would be able to maintain the lengths of the different seasons and thereby the seasonality, and would not increase water loss via enhanced transpiration. Some recent studies, however, indicated that regional or global mean temperature may continue to increase following zero carbon emissions (Gillet et al. 2011; Frölicher et al. 2013), although a number of studies showed that global mean surface temperatures would stay roughly constant for a couple of centuries after carbon emissions are stopped (Matthews and Caldeira 2008; Solomon et al. 2009). It also remains unclear when zero carbon emissions will be achieved. Therefore, the stabilization of phenology globally is not anticipated for the foreseeable future.”

Reviewer #3 (Remarks to the Author):

I approve of the revisions and authors responses to my previous concerns.

I did find this error in the final paragraph though: 'Rising air temperatures driven by the buildup of carbon dioxide and other

greenhouse gases can advance SOS and/or delay SOS and thereby lead to the lengthening of the growing

season'

I think that this should read 'SOS and/or delay of EOS' because it does not make sense currently.

ash

Response: We sincerely thank Reviewer #3 for the time and effort in reviewing our manuscript. We corrected the error in the revised manuscript (Line 817).

References

Matthews, H. D. & Caldeira, K. Stabilizing climate requires near-zero emissions. *Geophys. Res. Lett.* 35, L04705 (2008).

Solomon, S., Plattner, G. K., Knutti, R. & Friedlingstein, P. Irreversible climate change due to carbon dioxide emissions. *Proc. Natl Acad. Sci. USA* 106, 1704–1709 (2009).

Gillet, N. P., Arora, V. K., Zickfeld, K., Marshal, S. J. & Merryfield, W. J. Ongoing climate change following a complete cessation of carbon dioxide emissions. *Nature Geosci.* 4, 83–87 (2011).

Frölicher, T.L., Winter, M., Sarmiento, J.L. (2013) Continued global warming after CO2 emissions stoppage. *Nature Climate Change*, 4, 40-44.